



# Revisiting global satellite observations of stratospheric cirrus clouds

Ling Zou[1,2], Sabine Griessbach[2], Lars Hoffmann[2], Bing Gong[2], and Lunche Wang[1]

[1]Hubei Key Laboratory of Critical Zone Evolution, School of Geography and Information Engineering, China University of Geosciences, Wuhan, China
[2]Jülich Supercomputing Centre (JSC), Forschungszentrum Jülich, Jülich, Germany

**Correspondence:** Ling Zou (l.zou@fz-juelich.de; cheryl_zou@whu.edu.cn)

**Abstract.** As knowledge about the cirrus clouds in the lower stratosphere is limited, reliable long-term measurements are needed to assess their characteristics, radiative impact and important role in upper troposphere and lower stratosphere (UTLS) chemistry. To investigate the global and seasonal distribution of stratospheric cirrus clouds, we used the latest version (V4.x) of the Cloud-Aerosol Lidar and Infrared Pathfinder Satellite Observations (CALIPSO) and Michelson Interferometer for Passive

Atmospheric Sounding (MIPAS) data. For the identification of stratospheric cirrus clouds, precise information on both, the cloud top height (CTH) and the tropopause height is crucial. Here, we used lapse rate tropopause heights estimated from the ERA-Interim global reanalysis. Considering the uncertainties of the tropopause heights and the vertical sampling grid of the CALIPSO data, we considered cirrus clouds with CTHs more than 0.5 km above the tropopause as being stratospheric. We focused on nighttime CALIPSO measurements, because of their higher detection sensitivity. A six-year mean (2006 –

2012) global distribution of stratospheric cirrus cloud from CALIPSO showed that higher CTH occurrence frequencies are observed in the tropics than in the extra-tropics. Tropical hotspots of stratospheric cirrus clouds associated with deep convection are located over Equatorial Africa, South and Southeast Asia, the western Pacific and South America. Stratospheric cirrus clouds were more often detected in December-February (15 %) than June-August (8 %) in the tropics ($\pm$ 20°). At middle (40-60°) and higher latitudes (> 60°), CALIPSO observed on average about 2 % stratospheric cirrus clouds. Observations of

stratospheric cirrus cloud with MIPAS are presented here for the first time. Taking into account the MIPAS vertical sampling and broad field of view, we considered cirrus CTHs detected not less than 0.75 km above the tropopause as being stratospheric. Compared to CALIPSO, MIPAS observed twice as many stratospheric cirrus clouds at northern and southern middle latitudes (occurrence frequencies of 4 – 5 % for MIPAS rather than about 2 % for CALIPSO). We attribute more frequent observations of stratospheric cirrus clouds with MIPAS to higher detection sensitivity of the instrument to optically thin clouds. Sensitivity

tests on MIPAS stratospheric cloud detections have been conducted to rule out sampling artefacts. Future work should focus on better understanding the origin of the stratospheric cirrus clouds and their impact on radiative forcing and climate.



## 1 Introduction

Cirrus clouds are optically thin ice clouds that form at cold temperatures in the middle and upper troposphere. They cover roughly about $20 - 40\%$ of the globe (Liou, 1986; Wylie et al., 1994, 2005). As of their wide coverage and high occurrence frequencies, cirrus clouds play an important role in changing the surface energy budget of the earth-atmosphere system (Berry and Mace, 2014; Hong et al., 2016), affecting the distribution of water vapor and thermal structure of the atmosphere (Schoeberl et al., 2019), and influencing the climate (Corti et al., 2006; Schoeberl and Dessler, 2011; Dinh et al., 2012; Dessler et al., 2016). The characteristics and distribution of cirrus clouds are among the most sensitive parameters to climate variability.

To better understand the formation, evolution, and climate effects of cirrus clouds, the exploration of their global geospatial distribution and occurrence frequencies is essential. Depending on the satellite instruments sensitivities, the derived occurrence frequencies significantly differ, e. g. in global average $34.9\%$ cirrus clouds were observed by the High Resolution Infrared Radiometer Sounder (HIRS) between June 1989 to May 1993 (Wylie et al., 1994), $16.7\%$ were derived from a joint analysis of the space-borne cloud radar (CloudSat) and the Cloud-Aerosol Lidar and Infrared Pathfinder Satellite Observations (CALIPSO) for the period from June 2006 to June 2007 (Sassen et al., 2008), and $13.5\%$ are reported in the International Satellite Cloud Climatology Project (ISCCP) D2 data that was acquired between 1984 and 2004 by nadir viewing satellite instruments (Eleftheratos et al., 2007). More observations from additional resources are therefore urgently required to clarify the global occurrence of cirrus clouds in a changing climate.

Despite the differences in the global occurrence frequencies, some consistencies with respect to the spatial and temporal distribution of cirrus clouds can be seen between the studies. For instance, cirrus clouds occur more often in the tropics than in the extra-tropics (Wang et al., 1996; Nazaryan et al., 2008). Another general agreement on the geospatial distribution of tropical cirrus clouds is that high occurrence frequencies are generally detected over Equatorial Africa, South and Southeast Asia, the western Pacific and South America (Riihimaki and McFarlane, 2010; Massie et al., 2013). The largest occurrence frequencies of tropical cirrus clouds generally occur in boreal winter and minimum frequencies appear in boreal summer (Massie et al., 2010; Wang and Dessler, 2012). Considering the vertical distribution of the cloud fraction, Fu et al. (2007) found about $0.05\%$ at $18.5\,km$, $0.5\%$ at $18.0\,km$ and $5\%$ at $17.0\,km$ between $20°\,S$ and $20°\,N$ from CALIPSO observations, which indicated the occurrence of cirrus clouds in the lower stratosphere. Dessler (2009) was the first to analyze the occurrence of cirrus cloud in the lower stratosphere with CALIPSO measurements in the Northern Hemisphere. The impact of stratospheric cirrus clouds on climate variability is still unclear and studies on the occurrence of stratospheric cirrus clouds are still limited and controversial.

Stratospheric cirrus clouds have been reported in the tropics and at middle latitudes from in situ, ground based lidar, and satellite measurements. Studies of stratospheric cirrus clouds from in situ measurements are rare for the tropics (De Reus et al., 2009), middle latitudes (Clodman, 1957) and high latitudes (Lelieveld et al., 1999; Kärcher and Solomon, 1999). Reports of the appearance of stratospheric cirrus clouds from ground-based lidar measurements are provided more often at middle latitudes (Goldfarb et al., 2001; Keckhut et al., 2005; Noël and Haeffelin, 2007; Rolf, 2012) and in the tropics (Sivakumar et al., 2003; Sandhya et al., 2015). Among the satellite instruments, CALIPSO (Dessler, 2009; Pan and Munchak, 2011; Iwasaki et al., 2015) and Cryogenic Infrared Spectrometers and Telescopes for the Atmosphere (CRISTA) (Spang et al., 2015) were used





to investigate stratospheric cirrus clouds. The distribution of stratospheric cirrus clouds in the tropics follows the general distribution of cirrus clouds, highest fractions being found over Equatorial Africa, South and Southeast Asia, the western Pacific and South America. However, the consistency and agreement on the occurrence are still under debate as the results varied measurement-by-measurement and even study-by-study based on the same instrument.

At middle latitudes, Noël and Haeffelin (2007) found ∼2.5 % of cirrus clouds above the first tropopause based on ground-based lidar measurements over France. Pan and Munchak (2011) noted ∼2 % cirrus clouds with cloud top heights (CTHs) 0.5 km above the tropopause in the northern hemisphere when using tropopause heights derived from the National Centers for Environmental Prediction Global Forecast System (GFS). In another study, about twice as many stratospheric cirrus clouds (∼5 %) were detected during two weeks of infrared limb emission measurements in boreal summer 1997 of CRISTA instrument

over 40°-60° N (Spang et al., 2015). Potential reasons for these differences would be the higher detection sensitivity of IR limb emission measurements compared to the standard CALIPSO data product and a sampling bias due to the comparison of a 2 week period in 1996 versus a four-year three-monthly mean between 2006 and 2010. Further measurements with high detection sensitivity to stratospheric cirrus are indispensable.

Investigations of stratospheric cirrus clouds including high latitudes (> 60°) are rare (Pan and Munchak, 2011; Spang et al.,

2015). The statistical values for the high latitude stratospheric cirrus clouds are with large uncertainty, which may be on account of the low detection sensitivity, coarse classification accuracy between polar stratospheric clouds and stratospheric cirrus clouds (Sassen et al., 2008) and tropopause uncertainties at polar latitudes (Zängl and Hoinka, 2001). Therefore, high detection accuracy and sensitivity measurements are of significant importance for investigating the global occurrence and distribution of stratospheric cirrus clouds.

In this study, we are revisiting and exploring the global features of stratospheric cirrus clouds with the high vertical resolution space lidar CALIPSO and the high sensitivity mid-infrared limb emission sounder MIPAS. The CALIPSO and MIPAS instruments, the stratospheric cirrus cloud top heights derived from the instruments, and the tropopause data used in this study are described in Sect. 2. As MIPAS and CALIPSO measurements have an overlap between June 2006 and April 2012, seasonal CTH occurrence frequencies of stratospheric cirrus during that time period are presented for CALIPSO in Sect. 3 and MIPAS

in Sect. 4. Since the comparison between day- and nighttime CALIPSO measurements showed that the nighttime measurements are more suitable for thin cirrus cloud detection (Sect. 3), the comparisons between MIPAS and CALIPSO occurrence frequencies of CTHs relative to the tropopause and for seasonal occurrence frequencies were only performed for nighttime measurements (Sect. 4). A comparison to four years (2006-2010) of stratospheric cirrus cloud statistics investigated by Pan and Munchak (2011) using an earlier version of CALIPSO data is presented in Sect. 5. Conclusions of this study are drawn in

Sect. 6.



## 2 Data sets

### 2.1 CALIPSO

The CALIPSO satellite (Winker et al., 2007, 2009) was launched on 28 April 2006 as a member of the afternoon constellation (A-Train) satellite constellation. In September 2018, CALIPSO exited the A-Train and joined CloudSat to be a part of the C-Train. The Cloud-Aerosol Lidar with Orthogonal Polarization (CALIOP) is a two-wavelength polarization-sensitive lidar instrument on CALIPSO. It probes the high-resolution vertical structure and properties of clouds and aerosols on a near global scale. The vertical resolution of CALIPSO is 30 m from 0.5 to 8.2 km, 60 m from 8.2 to 20.2 km and 180 m from 20.2 to 30.1 km. Studies found 96.3 % estimation accuracy of the CALIOP sensor for characterizing the cloud cover compared with the Moderate Resolution Imaging Spectroradiometer (MODIS) and CloudSat Cloud Profiling Radar (CPR) (Chan and Comiso, 2013). CALIPSO is suitable for high altitude cirrus clouds detection (Davis et al., 2010). The Vertical Feature Mask data (CAL_LID_L2_VFMStandardV4) used in this study were generated with a new set of cloud-aerosol discrimination (CAD) probability distribution functions. The increased spatial resolution provided an overall improvement in CAD reliability (Liu et al., 2019). Cirrus clouds and deep convective clouds are identified by the Feature Classification Flags based on the CALIPSO CAD algorithm as well as the International Satellite Cloud Climatology Project (ISCCP) definitions. To ensure a high confidence level of the data, only cirrus and deep convective clouds that are marked with a high feature type quality were extracted and analyzed in our study. Furthermore, day- and nighttime data were flagged and analyzed separately to take into account the different detection sensitivities.

For data processing, we first analyzed the vertical structure of all cirrus and deep convective clouds reported in the CALIPSO Vertical Feature Mask data. For multi-layer profiles, layers were combined if their vertical distances are less than 120 m (Martins et al., 2011). As we are interested in cirrus clouds in the upper troposphere and lower stratosphere region, co-located tropopause data are used to limit the analysis to CTHs in the range of ±4 km around the tropopause. An additional filter for polar stratospheric clouds (PSC) for high latitudes is indispensable as PSCs are identified as cirrus clouds by the CALIPSO classification algorithm. The PSC filter follows the criteria of Sassen et al. (2008), i. e., cloud layers were excluded if CTHs were higher than 12.0 km poleward of 60° N and 60° S during local winter time. The CTH occurrence frequency is defined as the ratio of the number of cirrus cloud top height detections to total number of profiles in a given region. Two examples of nighttime stratospheric cirrus and Antarctic PSCs are shown in Fig. 1. Those two stratospheric cirrus cases are detected in the tropics and at middle latitudes, respectively, and they are both associated with deep convection. In the tropics, the tops of the clouds reach up to 18 km, and the tops of clouds at middle latitudes reach up to 12.5 km which are 500m above the tropopause. The PSCs over Antarctic are excluded in our study as their cloud tops are more than 4 km above the tropopause.



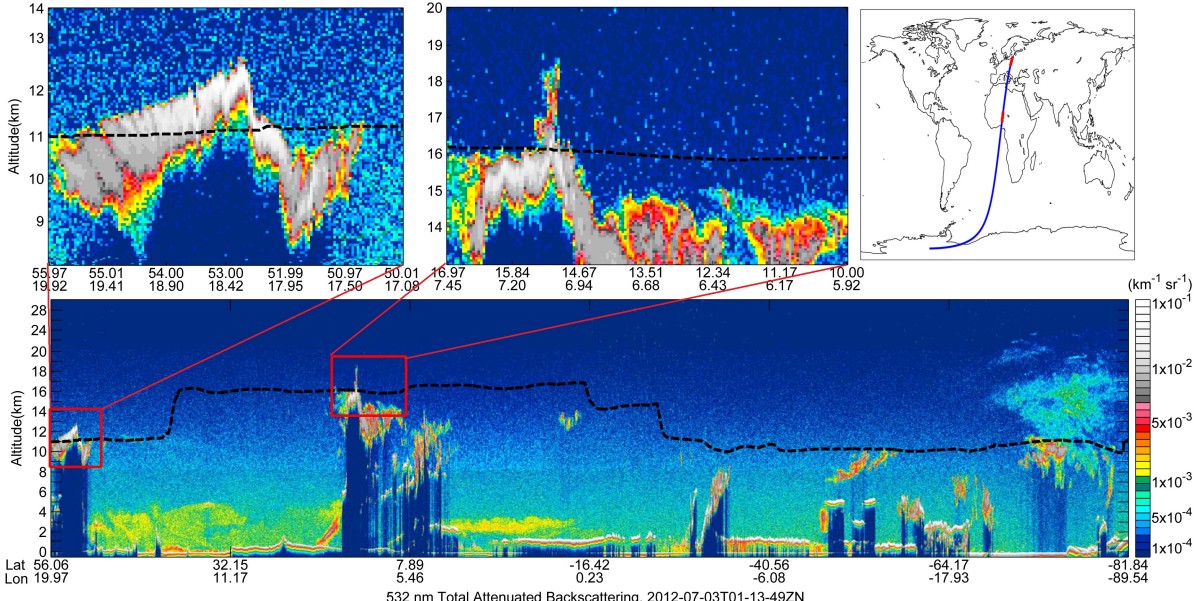

**Figure 1.** CALIPSO total attenuated backscatter at 523 nm observed on 3 July 2012 just after 01:13 UTC. The dashed black line indicates the lapse rate tropopause as estimated from the ERA-Interim reanalysis.

## 2.2 MIPAS

The Michelson Interferometer for Passive Atmospheric Sounding (MIPAS) onboard ESA's Envisat is a Fourier transform spectrometer for the detection of high-resolution limb emission spectra from the mid troposphere to the mesosphere (Fischer et al., 2008). MIPAS measured from July 2002 to April 2012 at a local solar time of 10:00 and 22:00 for the ascending and descending node, respectively. The field-of-view provides a resolution of 3 km (vertical) × 30 km (horizontal) at the tangent point. After January 2005, the vertical sampling below 21 km in nominal measurement mode was optimized to 1.5 km. The detectors cover the spectral range from 685 $cm^{-1}$ to 2410 $cm^{-1}$. In this work, the band A (685-980 $cm^{-1}$) and band B (1205-1510 $cm^{-1}$) version 8.03 level 1b data were used to derive cirrus cloud top heights.

The cirrus cloud detection was performed in two steps. First, the cloud detection was performed using the aerosol cloud index (ACI) (Griessbach et al., 2016). The ACI is defined as the maximum value of the cloud index (CI) and the aerosol index (AI). The CI is the ratio of the mean radiances of a strong $CO_2$ emission band [788.25, 796.25 $cm^{-1}$] and an atmospheric window band [832.31, 834.37 $cm^{-1}$] (Spang et al., 2001a, b). The AI is defined as the ratio of the mean radiances in the same $CO_2$ emission band [788.25, 796.25 $cm^{-1}$] and another atmospheric window band [960.00, 961.00 $cm^{-1}$]. We used an ACI threshold of 7 to separate between clear air (ACI > 7) and cloudy air (ACI <= 7), as this value provides comparable results to the most sensitive altitude and latitude variable thresholds for the CI (Sembhi et al., 2012; Griessbach et al., 2016). In the second step, we filtered out aerosol from the detected clouds by applying the volcanic ash detection method of Griessbach et al. (2014)





and a brightness temperature difference correlation method that separates volcanic ash, mineral dust, and sulfate aerosol from ice clouds (Griessbach et al., 2016).

In this study, the top most tangent height of the ice cloud detection was extracted as cloud height to analyze stratospheric cirrus clouds with MIPAS. However, one shortcoming of the MIPAS measurements is the coarser vertical resolution and large

field-of-view. The large field-of-view, broken cloud conditions, and different extinction coefficients of the cloud layers cause CTHs uncertainties for MIPAS. For optically thick clouds, CTHs can be overestimated up to ∼1.6 km due to the field-of-view, and for optically thin clouds CTH can be underestimated up to ∼5.1 km (Griessbach et al., 2020). An average CTH overestimation of 0.75 to 1 km compared to HIRDLS and CALIPSO has been reported by Sembhi et al. (2012). Therefore, sensitivity tests of CTHs in MIPAS are indispensable to assess the robustness of the results. After extracting the cirrus cloud

heights, we applied the same PSC filter as for CALIPSO. Further, day and night time flags for MIPAS were added based on the solar zenith angles of the observations.

## 2.3 Tropopause data

The lapse rate tropopause (LRT) is defined as the lowest level at which the lapse rate decreases to 2° C/km or less, provided the average lapse rate between this level and all higher levels within 2 km does not exceed 2° C/km (WMO, 1957). Due to the close

relations to temperature and relative humidity, the LRT shows good agreement with sharp stability and chemical transitions between the troposphere and stratosphere, globally (Pan and Munchak, 2011; Spang et al., 2015; Xian and Homeyer, 2019). The LRT is therefore considered crucial for stratospheric cirrus cloud detections (Spang et al., 2015). In this study, we used LRT geopotential heights derived from the ERA-Interim reanalysis (re3data.org, 2020). ERA-Interim is a global atmospheric reanalysis with approximately 0.75° grid resolution on 60 vertical levels from the surface up to 0.1 hPa, which is available

6-hourly from 1979 to August 2019 (Dee et al., 2011). Considering a typical ±0.3 km bias of the ERA-Interim LRT data with respect to Global Positioning System (GPS) measurements and the 0.2 km vertical grid sampling of the CALIPSO data, an uncertainty of 0.5 km was used for stratospheric cirrus cloud detections, which is comparable to the approach of Homeyer et al. (2010) and Pan and Munchak (2011). The term "stratospheric cirrus clouds" for CALIPSO hereafter indicates cirrus clouds that have CTHs being at least 0.5 km above the tropopause.

## 155 3 Stratospheric cirrus clouds measured by CALIPSO from 2006-2012

### 3.1 Night- and daytime stratospheric cirrus clouds

The CALIPSO night- and daytime mean stratospheric cirrus cloud fractions are presented in Fig. 2. Although similar patterns are observed in the tropics, 2-3 times higher frequencies of stratospheric cirrus clouds are detected at nighttime than at daytime. The highest fraction at nighttime is located over Central Africa with a maximum of ∼0.36, whereas it is <0.16 at daytime.

The regional mean CTH occurrence frequency of stratospheric cirrus clouds in the tropics is ∼10 % at nighttime (Fig. 2c) but ∼4 % at daytime (Fig. 2d). At middle and high latitudes, there are rare stratospheric cirrus cloud detections at daytime. The



regional mean fractions over middle latitudes in the southern and northern hemisphere are ∼1 % during the daytime and ∼2 % during the nighttime. The sensitivity of CALIPSO is by a factor of ∼2.5 at 18 km, ∼2 at 15 km, and ∼1.5 at 10 km higher at nighttime compared to daytime due to a better signal-to-noise ratio (Winker et al., 2009), which is in line with findings of Hunt

et al. (2009) and Getzewich et al. (2018). As high altitude cirrus clouds show little diurnal cycle and thin cirrus in particular do not show any diurnal pattern (Wylie et al., 1994), we consider the difference in detection sensitivity the leading cause for the difference between CALIPSO night- and daytime measurements. Hence, only nighttime CALIPSO measurements will be further analyzed in this study.

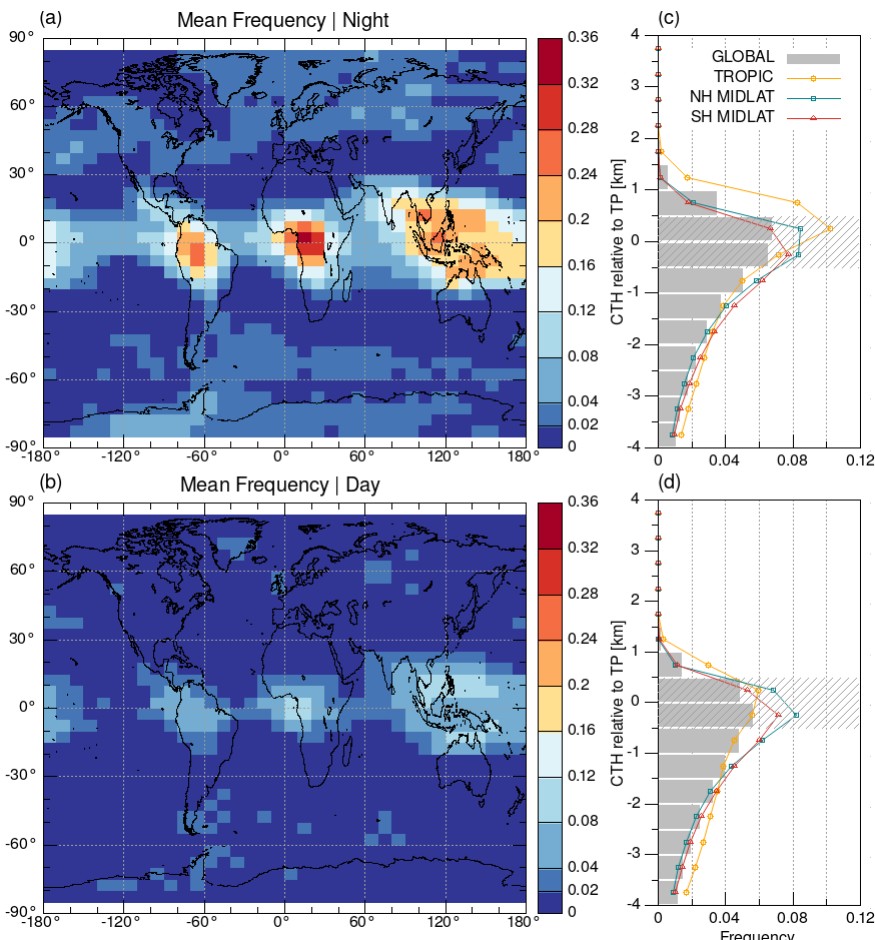

**Figure 2.** Global distribution of CTH occurrence frequencies of stratospheric cirrus clouds for June 2006 to April 2012 derived from CALIPSO a) nighttime and b) daytime measurements. The maps are shown on a 5°×10° latitude-longitude grid. The corresponding vertical CTH fraction profiles for c) nighttime and d) daytime are relative to the tropopause and show zonal means for the tropics [20° S, 20° N], northern middle latitudes (NH MIDLAT) [40° N, 60° N] and southern middle latitudes (SH MIDLAT) [40° S, 60° S]. The uncertainty of the tropopause is ±0.5 km and marked by the gray hatched area.





## 3.2 Seasonal nighttime stratospheric cirrus clouds

Seasonal geospatial distributions of nighttime stratospheric cirrus clouds are presented in Fig. 3 and seasonal vertical fractions of CTHs relative to the tropopause are shown in Fig. 4. The CTH occurrence frequencies of stratospheric cirrus clouds are globally similar for the four seasons with a maximum frequency of $\sim$5 % in DJF (December to February) and a minimum frequency of $\sim$4 % in SON (September to November). Regionally, high occurrence frequencies are observed in the tropics during all seasons over Equatorial Africa, South and Southeast Asia, the western Pacific and South America. The distribution

of stratospheric cirrus cloud hotspots in the tropics [20° S, 20° N] is consistent with the cirrus cloud hotspots reported by Wang et al. (1996) and Wylie et al. (2005). The seasonal tropical mean frequencies are in the range of $\sim$8 % to $\sim$15 % (Fig. 4) and are nearly 4 to 5 times higher than the middle latitude seasonal means.

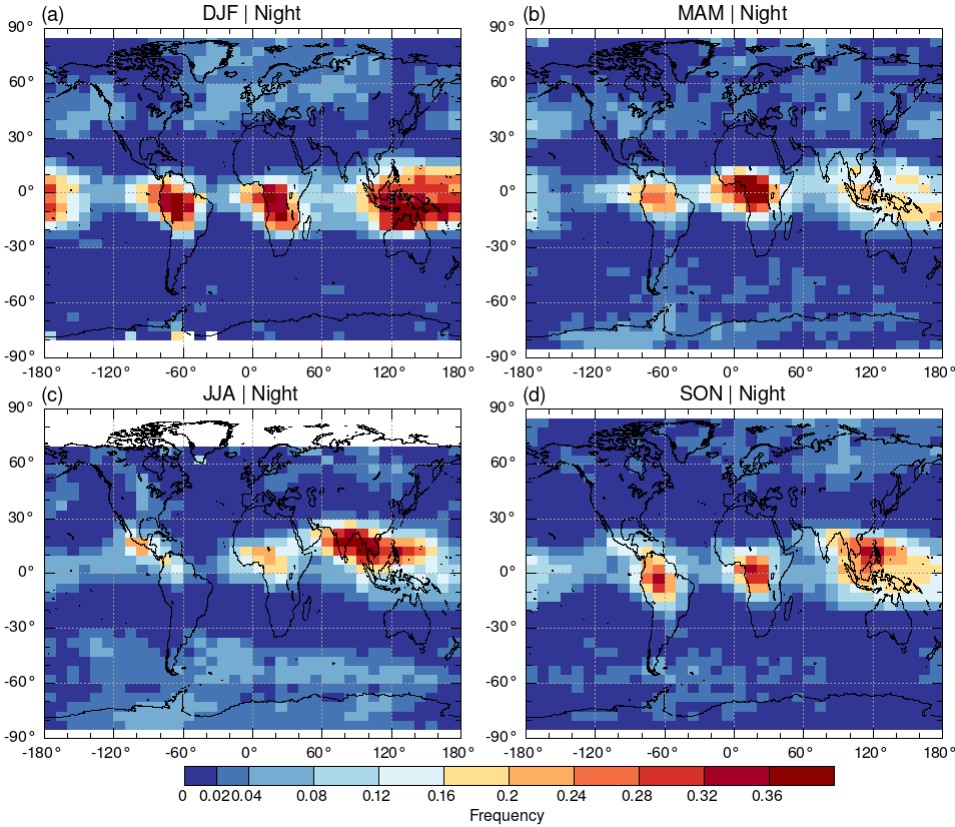

**Figure 3.** Seasonal CTH occurrence frequencies of nighttime stratospheric cirrus clouds derived from CALIPSO observations during June 2006 to April 2012 in a) December to February (DJF), b) March to May (MAM), c) June to August (JJA), and d) September to November (SON). The grid boxes are the same as in Fig. 2.



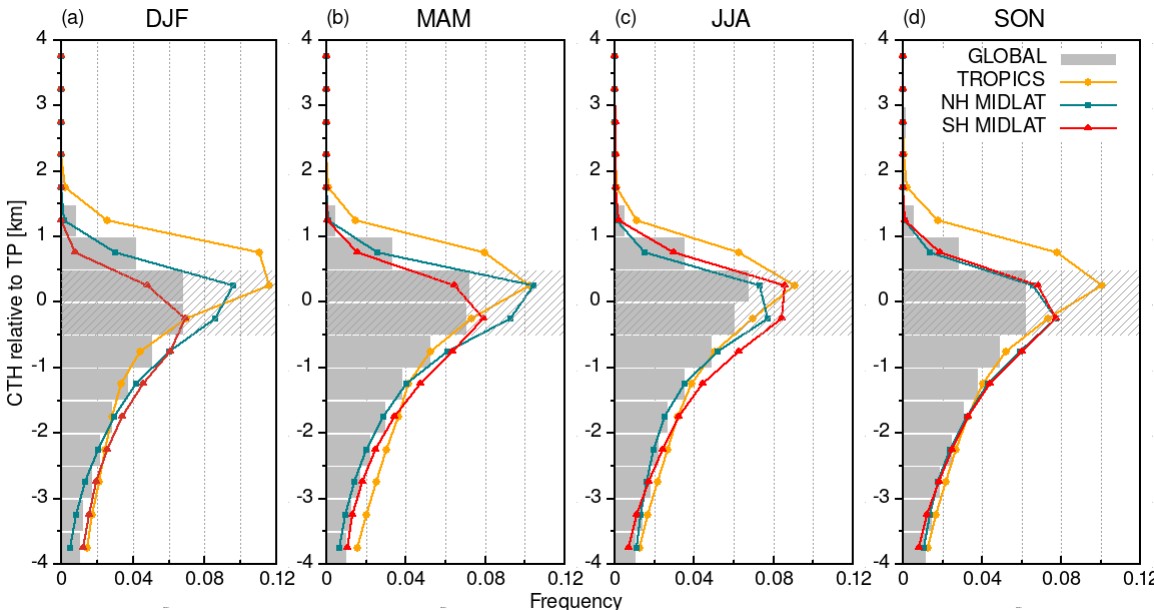

**Figure 4.** Vertical occurrence frequencies of nighttime CTHs relative to the tropopause derived from CALIPSO observations for the four seasons a) DJF, b) MAM, c) JJA, d) SON and the same time period as in Fig. 3.

In DJF, high frequencies of stratospheric cirrus clouds are mainly located south of the equator over Equatorial Africa, South and Southeast Asia, the western Pacific, and South America with highest fractions up to 0.36 (Fig. 3a). The tropical mean
frequency is ∼0.15 (Fig. 4a). Rare stratospheric cirrus clouds are observed in southern hemisphere middle and high latitudes, while 4-8 % stratospheric cirrus clouds are found over western North America, the North Atlantic, Europe and northern Asia (Fig. 3a). The regional mean frequencies for southern middle latitudes [40° S, 60° S] and northern middle latitudes [40° N, 60° N] are 1 % and 3 %, respectively (Fig. 4a).

In MAM (March to May), the tropical hotspots show slightly northward movement following the intertropical convergence
zone (ITCZ) and are mainly located over Equatorial Africa (Fig. 3b). Significantly more stratospheric cirrus clouds are present at southern high latitudes and the frequency at southern middle latitudes increases to ∼2 % (Fig. 4b).

In JJA (June to August), stratospheric cirrus clouds in the tropics are mainly located in the deep convection regions of the ITCZ that are now north of the equator over Middle America, southern Asia, southern India and the Bay of Bengal (Fig. 3c). The regional mean frequency for the tropics in JJA (Fig. 4c) may be slightly underestimated as the highest frequency is
located at 20° N. Many stratospheric cirrus clouds are detected over southern middle and high latitudes during this time. 4-8 % stratospheric cirrus clouds show up over central northern America and northern Asia, but observations are missing at northern high latitudes due to the satellite orbit (Fig. 3c). In the oceanic downwind region of the southern tip of South America a band with 4-8 % stratospheric cirrus cloud observations is visible (Fig. 3c). The regional mean frequency for southern and northern middle latitudes are 3.1 % and 1.8 % in JJA (Fig. 4c).





In SON the hotspots of stratospheric cirrus clouds in the tropics are located between [20° S, 20° N] with maximum frequencies not exceeding 36 % (Fig. 3d) and a mean frequency of about 10 % (Fig. 4d). Similar frequencies are found over the middle and high latitudes of both hemispheres. The frequencies at the middle and high latitudes of both hemispheres are comparable and mostly below 4 % (Fig. 3d). On average the stratospheric cirrus cloud occurrence frequencies are $\sim$2 % at northern and southern middle latitudes (Fig. 4d).

The seasonal shifts of the hotspots in the tropics perfectly match the location of high convective frequencies and of the overshooting precipitation features that are following the ITCZ (Schoeberl et al., 2019). Highest occurrence frequencies are observed south of the equator in DJF and north of the equator in JJA, which is in agreement with the seasonal distribution of high cirrus clouds (Wang et al., 1996; Iwasaki et al., 2015). Although the occurrence frequencies at middle latitudes are lower compared to the tropics, we see higher occurrence frequencies during the winter months.

## 4 Stratospheric cirrus clouds measured by MIPAS from 2006-2012

### 4.1 Nighttime cloud top height occurrence frequencies in the UTLS

The CALIPSO and MIPAS occurrence frequencies of CTHs relative to the tropopause are compared globally, seasonally, and latitudinally resolved in Fig. 5. The analysis is restricted to nighttime measurements, because of the higher detection sensitivity of CALIPSO at nighttime (shown in Sect. 3.1). At all altitudes within the range of ±4 km around the tropopause, cirrus CTH
fractions from MIPAS and CALIPSO show similar vertical distributions with the highest frequencies around the tropopause. A maximum of cirrus cloud top height occurrences around the tropopause is also reported in other studies relying on CALIPSO (e.g., Pan and Munchak, 2011; He et al., 2013) and ground based lidar data (Goldfarb et al., 2001; Sivakumar et al., 2003; Seifert et al., 2007; Noël and Haeffelin, 2007).

 In most cases, MIPAS detects more cirrus clouds than CALIPSO, resulting in 2 percentage points (pp) more cirrus cloud
detections for the all-year mean. The reasons for the generally higher frequencies observed by MIPAS are a) a higher detection sensitivity towards optically thin cirrus clouds as its detection sensitivity goes down to optical depths ($\tau$) of $10^{-5}$ (Sembhi et al., 2012) whereas the minimum optical depth for CALIPSO is about $10^{-3}$ (Martins et al., 2011) and b) the long line of sight, which samples about 200 km around the tangent point, that makes MIPAS more likely to sample a cloud than the CALIPSO nadir measurements. Differences due to the diurnal cycle we consider negligible as the CTH occurrence frequencies
of high altitude cirrus clouds in many cases are constant or even show a slight increase from 10:00 pm (MIPAS local equator overpath time) to 1:30 am (CALIPSO local equator overpath time) (Noel et al., 2018, Fig. 5).

 The absolute differences between MIPAS and CALIPSO CTH occurrence frequencies show two maxima above and below the tropopause. Only in the southern hemisphere middle latitudes, the maximum below the tropopause is missing in DJF and MAM. On global average both maxima are comparable, but it varies depending on season and latitude. Regionally, in DJF and
MAM the maximum in the stratosphere is dominating and in JJA the maximum in the troposphere is more pronounced. The maximum differences at altitudes of 500 m above the tropopause are 3.3, 3.5, and 4.2 pp and the average differences are 1.3,





1.2, and 1.7 pp in the tropics, southern, and northern middle latitudes, respectively. The minimum difference is located at the tropopause and reaches even zero in the tropics.

**Figure 5.** Global and regional mean occurrence frequencies of CTHs relative to the tropopause from nighttime measurements. The red bars indicate the MIPAS measurements and blue bars indicate the CALIPSO measurements. The green dotted lines are the differences between MIPAS and CALIPSO measurements. The first column shows the yearly mean values and the other four columns are values for the four seasons DJF, MAM, JJA, and SON. The rows a, b, c, d present the average values over the globe, tropics, northern middle latitudes, and southern middle latitudes.





MIPAS cloud measurements are known to overestimate cloud top heights by 0.75-1 km on average compared to CALIPSO
(Sembhi et al., 2012), but the profiles in Fig. 5 do not exhibit any obvious altitude shift. In a recent study, Griessbach et al. (2020) showed that the uncertainty of the MIPAS cloud top heights depends on the cloud's optical thickness. For optically thick clouds ($0.3 < \tau < 3.0$), MIPAS' altitude error is between $-0.1$ and $1.6$ km (with 0.75 km on average). Whereas for subvisible cirrus clouds within CALIPSO's detection sensitivity range ($0.001 < \tau < 0.03$), the MIPAS altitude error would be lower, between about -0.65 and 0.5 km on average.

## 235  4.2  Stratospheric cirrus clouds

Although a systematic cloud top height overestimation was not immediately visible in comparison to CALIPSO in Fig. 5, it is the largest challenge for the detection of stratospheric cirrus clouds with MIPAS. The most conservative approach to derive stratospheric cirrus clouds from MIPAS data would be counting only clouds with CTHs 1.6 km above the tropopause, because this is the maximum possible overestimation for optically thick clouds due to MIPAS' field-of-view and vertical
sampling (Griessbach et al., 2020). In practice, for the optically thickest clouds the CTH uncertainty ranges from -0.1 to 1.6 km (Griessbach et al., 2020). Assuming that in our nearly 6 years of statistics the tangent heights are equally distributed with respect to the cloud top, we expected an average overestimation of 0.75 km. This value is in agreement with an average overestimation of 0.75 km derived from a comparison between MIPAS and CALIPSO measurements of 3-months averages of a summer and a winter season (Sembhi et al., 2012).

**Table 1.** Regional mean CTH occurrence frequencies of stratospheric cirrus clouds from CALIPSO and MIPAS measurements.

| Instrument (CTHs detection thresholds) | TROPICS [30° S-30° N] | NH MIDLAT [40° N-60° N] | SH MIDLAT [40° S-60° S] |
|---|---|---|---|
| CALIPSO (CTHs > 0.50 km) | 0.073 | 0.022 | 0.019 |
| MIPAS (CTHs > 0.65 km) | 0.091 | 0.052 | 0.064 |
| MIPAS (CTHs > 0.70 km) | 0.084 | 0.046 | 0.058 |
| MIPAS (CTHs > 0.75 km) | 0.077 | 0.040 | 0.052 |
| MIPAS (CTHs > 0.80 km) | 0.070 | 0.034 | 0.046 |
| MIPAS (CTHs > 0.85 km) | 0.064 | 0.030 | 0.041 |
| MIPAS (CTHs > 0.90 km) | 0.058 | 0.025 | 0.035 |
| MIPAS (CTHs > 0.75 km, MD-TP > 1.10 km) | 0.077 | 0.037 | 0.047 |
| MIPAS (CTHs > 0.75 km, MD-TP > 1.15 km) | 0.074 | 0.032 | 0.040 |
| MIPAS (CTHs > 0.75 km, MD-TP > 1.20 km) | 0.066 | 0.027 | 0.034 |
| MIPAS (CTHs > 0.75 km, MD-TP > 1.30 km) | 0.049 | 0.020 | 0.023 |

Here, we made the assumption that stratospheric cirrus clouds in the tropics have optical thicknesses that are detectable by CALIPSO. For fresh overshooting convection events the optical thickness is above CALIPSO's detection limit (e.g. De Reus et al., 2009). However, for subvisual cirrus clouds CALIPSO was estimated to miss up to 66 % (Davis et al., 2010).



An analysis of cloud occurrence frequencies of three years of Optical Spectrograph and InfraRed Imager System (OSIRIS)
measurements showed that in the tropics on average about 13 % of the clouds between 12 and 25 km have an optical thickness
below $5 \times 10^{-3}$ (Bourassa et al., 2005). Since CALIPSO's lower ice cloud detection limit is about $1 \times 10^{-3}$, we assume that on
average a small amount of subvisible cirrus clouds in the tropics will be missed by CALIPSO. However, our main goal was
to derive information on middle and high latitude stratospheric cirrus clouds and hence, we accepted an underestimation of
tropical stratospheric cirrus and determined the optimal minimum distance to the tropopause for the MIPAS cloud detections
that minimizes the differences between MIPAS and CALIPSO stratospheric cloud occurrences in the tropics (Fig. 6a and
Tab. 1).

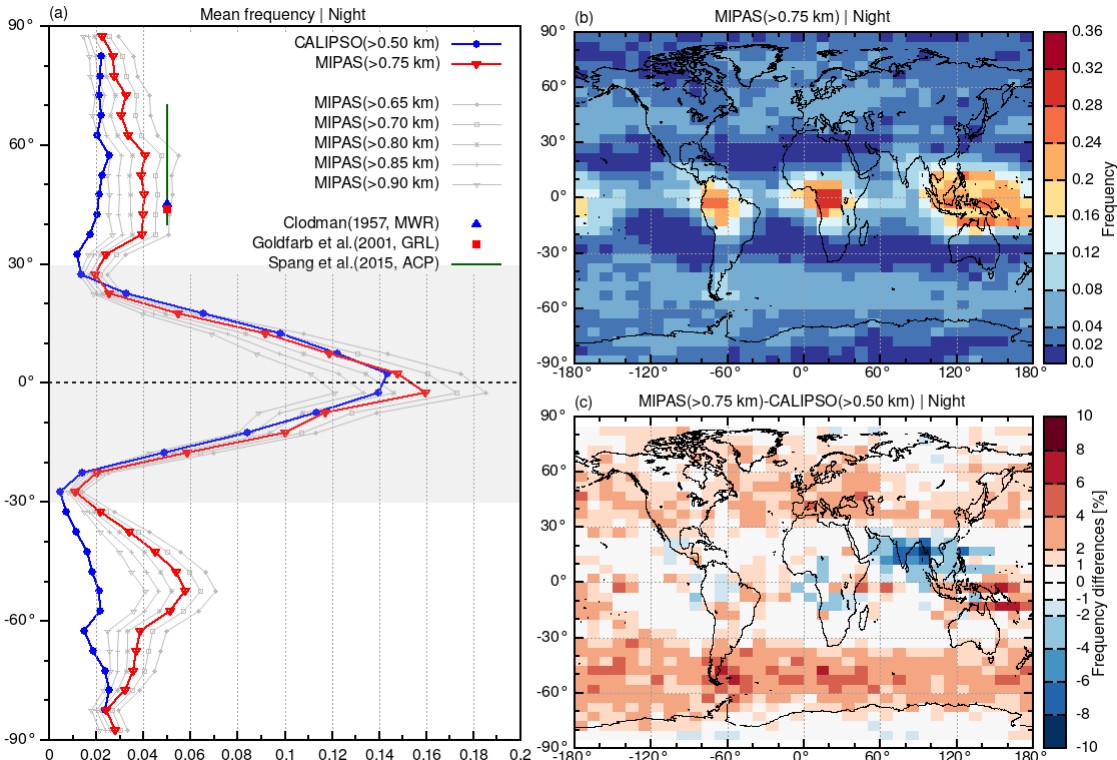

**Figure 6.** Zonal mean CTH occurrence frequencies (a) and geospatial distribution on a 5°×10° latitude-longitude grid (b) of 6-year mean
nighttime occurrence frequencies of stratospheric cirrus clouds observed by MIPAS (CTHs >0.75 km above tropopause). c) Difference
between MIPAS and CALIPSO (CTHs >0.5 km above tropopause) occurrence frequencies.


The minimum difference of stratospheric cirrus CTH frequencies in the tropics between CALIPSO (7.3 %) and MIPAS
(7.7 %) is 0.4 pp, when the minimum distance to the tropopause is 0.75 km for MIPAS (Fig. 6a and Tab. 1). Hence, we consider
CTHs 0.75 km above the tropopause as stratospheric clouds for MIPAS. With a 0.75 km tropopause threshold for MIPAS,
the CTH occurrence frequency of stratospheric cirrus clouds at northern middle latitudes is 4.0 % for MIPAS and ~2.2 % for
CALIPSO and at southern middle latitudes it is 5.2 % for MIPAS and 1.9 % for CALIPSO, respectively. MIPAS observed 1.8



to 2.6 times more stratospheric cirrus clouds at middle latitudes than CALIPSO, even though similar frequencies were found for the tropics.

The geospatial distribution of the CTH occurrence frequencies of stratospheric cirrus clouds observed by MIPAS and the differences to CALIPSO are presented in Fig. 6b and c. The general occurrence frequency patterns of both instruments are

rather similar (Fig. 2b), with hot spots in the tropics over Equatorial Africa, Southeast Asia, the western Pacific, and South America. However, significantly more stratospheric cirrus clouds are detected at middle and high latitudes by MIPAS. Although the average difference in the tropics is small, there are distinct patterns visible in the difference map (Fig. 6c). While MIPAS slightly underestimates the fractions of stratospheric cirrus clouds at the South American and Equatorial African hotspot, it overestimates the Southeast Asia/Western Pacific hotspot. The largest underestimation is found extending over the Indian

peninsula and the Bay of Bengal with a maximum difference of 6-8 pp. The seasonal geospatial distribution of stratospheric cirrus clouds in Fig. 7e-h shows that this underestimation is related to the Asian summer monsoon, whereas the underestimation over South America and Equatorial Africa occur in all seasons. The overestimation over Southeast Asia and the western Pacific mostly occurs in MAM.

As a possible cause for the higher occurrence frequencies found by MIPAS, we tested if a potentially non-sufficient aerosol

filtering could have caused the higher detection frequencies in MIPAS data. But, since we did not find any correlation with volcanic eruptions, which are the dominating source of MIPAS aerosol detections, we ruled this out. The average occurrence frequency of 4 % derived from MIPAS at northern hemisphere middle latitudes is closer to the occurrence frequencies that were derived from previous in situ, ground-based, and space-based measurements. From six years of aircraft-based measurements over Canada between 1950-1956, Clodman (1957) derived an occurrence frequency of approximately 5 % for stratospheric

cirrus clouds more than 2000 ft (0.61 km) above the tropopause. Despite the rather large measurement errors, Clodman (1957) considered this result "authentic". From about a week of space based CRISTA measurements in August 1997, Spang et al. (2015) also derived about 5 % stratospheric cirrus clouds at middle and high latitudes (up to 70°N) for CTHs more than 0.5 km above the tropopause. In lidar data measured between 1997 and 1999 at Haute Provence, France (43.9°N), Goldfarb et al. (2001) observed also 5 % clouds that had cloud top heights at least 1 km above the tropopause. We consider the higher

detection sensitivity of MIPAS towards thin clouds as the reason for the about 2 times higher CTH occurrences frequencies of stratospheric cirrus clouds at northern middle latitudes, 3 times larger frequencies at southern middle latitudes, and 1.5 pp larger frequencies at high latitudes in MIPAS measurements, which was already suggested by the comparison of the CTH occurrence frequencies around the tropopause in Fig.5. In the middle and high latitudes, MIPAS systematically observed more stratospheric clouds (Fig. 6c). In the southern hemisphere, the higher occurrence frequencies are in a band between about 35°

and 70° S and in the northern hemisphere they are more pronounced over the oceanic regions and Europe to western Russia. The higher occurrence frequencies in the middle latitudes show a seasonal dependence (Fig. 7a-d). During the summer months (JJA and DJF), the smallest cloud occurrence frequencies are present, which coincides with the generally observed pattern of high altitude clouds in climatologies (Rossow and Schiffer, 1999). The highest regional mean frequencies at southern and northern middle latitudes in MIPAS are observed in MAM with values of 5.5 % and 3.3 %, respectively, while it is ~2 % in





CALIPSO. In DJF, nearly 1 % middle and high latitude stratospheric cirrus clouds are detected in CALIPSO, but about 4% are detected by MIPAS.

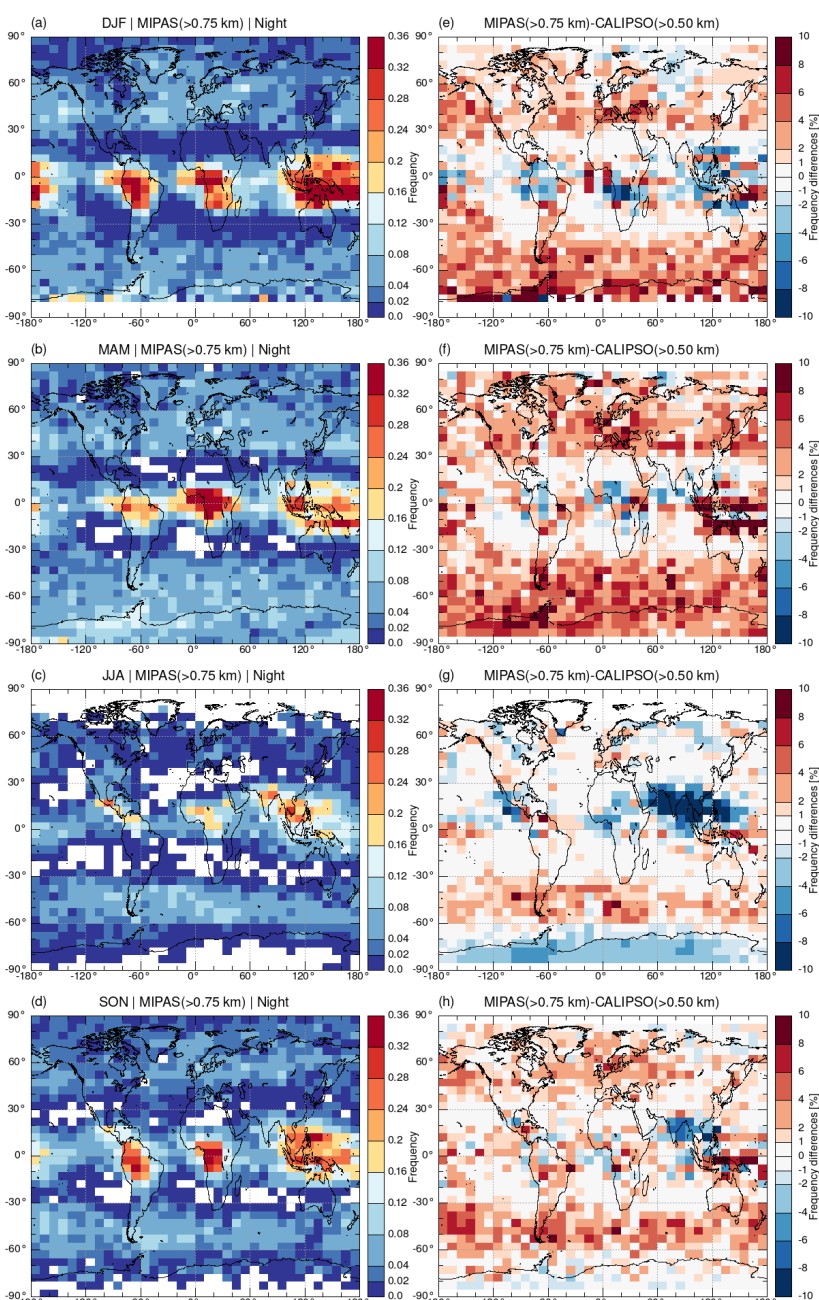

**Figure 7.** Seasonal nighttime mean CTH occurrence frequencies of stratospheric cirrus clouds observed by MIPAS (CTHs >0.75 km above tropopause) and the differences between MIPAS and CALIPSO (CTHs >0.50 km above tropopause). White boxes indicate that no stratospheric cirrus clouds were detected by MIPAS.



### 4.3 Diurnal cycle of cloud top height occurrences

The nighttime measurements of MIPAS and CALIPSO differ by about 3:30 h in equator crossing time ($\sim$10:00 pm and $\sim$1:30 am). High altitude cirrus clouds show little diurnal variation (Wylie et al., 1994). Over oceans the high altitude cloud occurrence measured by the Cloud-Aerosol Transport System (CATS) lidar is constant at middle latitudes and even slightly

increasing by up to 5 % between 30° N and 30° S (Noel et al., 2018, Fig. 6). Over land the behaviour is the same, except for SH middle latitudes, which is considered less significant due to the small amount of land masses there (Noel et al., 2018, Fig. 6). Differences of stratospheric cloud fractions measured by CATS at $\sim$10:00 pm and $\sim$1:30 am are less than 5 pp over equatorial Africa, South America and West Pacific in DJF and less than 2.5 pp over Central Africa and North Warm Pool (ocean) in JJA (Dauhut et al., 2020, Fig. 2). Ground based radar measurements in the United States Southern Great Plains show that the cloud

occurrence frequencies differ by less than 2 pp between 10:00 pm and 1:30 am (Zhao et al., 2017). Hence, the contribution of the diurnal cycle on cloud occurrence frequencies between CALIPSO and MIPAS is negligible.

Due to the same detection sensitivity of MIPAS for day- and nighttime measurements, we also analyzed the daytime data. The MIPAS nighttime and daytime stratospheric cirrus cloud statistics are compared in Fig. 8. The highest occurrence frequencies are observed in the tropics, where the daytime occurrence frequencies are about 1 pp smaller. At the middle latitudes the

daytime occurrence frequencies are slightly larger by less than 0.5 pp.

Assuming that stratospheric cirrus clouds correlate with high altitude cirrus clouds, result at middle latitudes is in agreement with the radar measurements above the United States Southern Great Plains where the all year mean cloud occurrence frequencies differ by less than 2 pp between 10:00 am and 10:00 pm (Zhao et al., 2017) and the CATS lidar measurements of high altitude cirrus clouds showing a deviation of less than 1 % from the daily mean in the middle latitudes, except in SH middle

latitudes over land (Noel et al., 2018).

In the tropics, the deviations of high altitude cirrus clouds from the all day mean between 10 am and 10 pm observed by CATS in JJA is up to 3 % over ocean and reaches up to about 18 % over land, where the larger occurrence frequencies are found during nighttime (Noel et al., 2018). Although these numbers appear large, 18 % of an average daily high altitude cloud occurrence frequency of 20 % (Figs. 2 and 3 in Noel et al., 2018) means an absolute difference of 3.6 pp. CATS daytime

data misses about 5 % of nighttime clouds due to a lower lidar sensitivity during daytime (Noel et al., 2018), which means a further reduction of the difference by 1 pp to 2.6 pp. Finally, the absolute difference of 2.6 pp between daytime and nighttime occurrence frequencies derived from CATS is valid for JJA, whereas the 1 pp difference between MIPAS daytime and nighttime occurrence frequencies is valid for the all year mean. A recent study on stratospheric cirrus cloud occurrences in the tropics derived from CATS measurements reports differences of about 3 to 10 pp in DJF and 5 to 7 pp in JJA between 10 am and 10 pm

Dauhut et al. (2020). This differs from our results that show only 1 pp difference between  10 am and 10 pm measurements. Although Noel et al. (2018) noted that CATS daytime measurements have to be averaged over 60 km to achieve a comparable detection sensitivity as the nighttime measurements averaged over 5 km, the results by Dauhut et al. (2020) were derived from 5 km along track averages at daytime and nighttime. The detection sensitivity of CATS measurements averaged over 5 km





during daytime is about 1.5 orders of magnitude lower than during nighttime (Yorks et al., 2016). Hence, we consider the
different detection sensitivities of CATS daytime and nighttime measurements as the main cause for the differences.

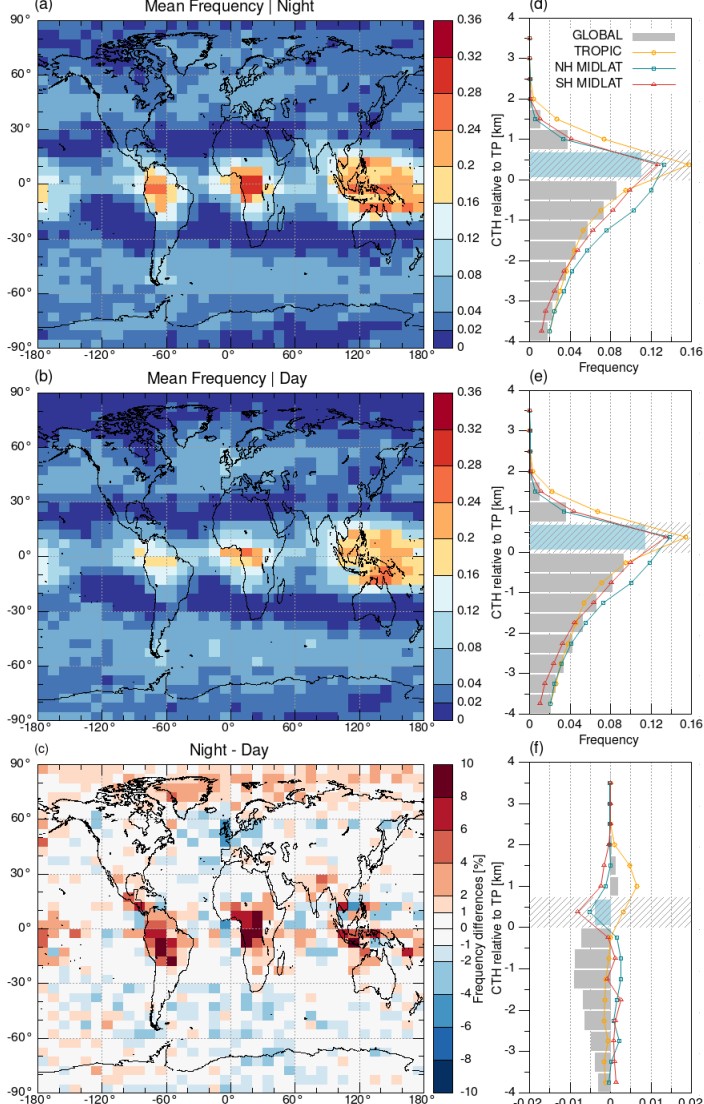

**Figure 8.** Nighttime (a) and daytime (b) CTH occurrence frequencies of stratospheric cirrus clouds derived from June 2006 to April 2012 MIPAS measurements. c) Difference between MIPAS nighttime and daytime occurrence frequencies. d), e), f) show the corresponding global mean frequencies of CTHs relative to the tropopause for nighttime, daytime, and their difference, respectively. The MIPAS tropopause threshold from 0 to 0.75 km above the tropopause is marked gray.





### 4.4 Sensitivity tests regarding the average distance to the tropopause

Figures 5 and 6 show that the occurrence frequencies of MIPAS and CALIPSO are closer to each other in the tropics than in the extra-tropics. To investigate potential sampling artifacts that arise from MIPAS sampling geometry, which approximately follows the tropopause, we calculated the mean of the distances of the CTHs of the stratospheric cirrus clouds to the tropopause

(MD-TP) in each grid box. Here again, only nighttime measurements were used. The means of the distances of the CTHs to the tropopause in Fig. 9 are larger in the tropics (1.1 to 1.3 km at the tropical hotspots) than at middle latitudes (0.75 to 1.0 km). Although these differences might relate to the 300 m low bias of the ERA-Interim tropopause heights in the tropics compared to GPS measurements and the different underlying causes for stratospheric cirrus clouds in the tropics, such as overshooting convection (De Reus et al., 2009; Iwasaki et al., 2015) and wave activity (Alexander et al., 2000), and in the extra-tropics, such

as double tropopause events (Noël and Haeffelin, 2007), we introduced an additional criterion for the MD-TP, so that it is more homogeneous at all latitudes to rule out sampling artefacts. To do so, we removed the lowest CTHs in each grid box until the required mean distance to the tropopause was reached, and hence we reduced the number stratospheric cirrus counts.

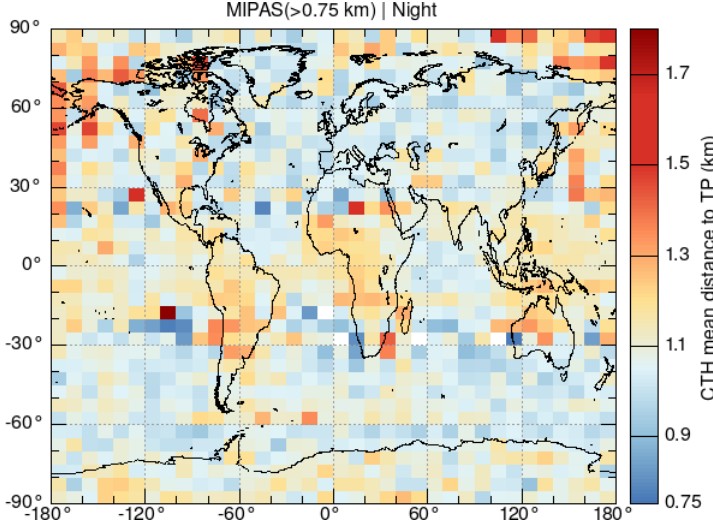

**Figure 9.** Mean distance of the MIPAS stratospheric cirrus CTHs to the tropopause in each grid box for the data shown in Fig. 6b.

Figure 10a and Table 1 show that with higher distance to the tropopause the zonal mean occurrence frequencies decrease. Again, we aimed for an optimal agreement between MIPAS and CALIPSO in the tropics, assuming that both instruments should

have similar detection capabilities here. The minimum difference between MIPAS and CALIPSO in the tropics was achieved (0.1 pp) for a MD-TP larger than 1.15 km. In this scenario MIPAS (CTHs >0.75 km and MD-TP>1.15 km), the CTH occurrence frequencies of stratospheric cirrus clouds are 3.2 % at northern hemisphere middle latitudes and 4.0 % at southern hemisphere middle latitudes. This is ∼0.5 to 0.7 pp smaller than for the statistics counting all clouds at 0.75 km above the tropopause, but still up to a factor of 2 larger than the CALIPSO occurrence frequencies. The overall stratospheric cloud occurrence patterns

in Fig. 10b remain the same as in Fig. 6b, but the positive differences in the extra-tropics are reduced and the already strong





negative difference related to the Asian summer monsoon got even stronger (compare Figs. 6c and 10c). Hence, we conclude that MIPAS' vertical sampling pattern is not the cause for the greater CTH occurrence frequencies detected at middle latitudes. We also conclude that the finding of higher CTH occurrence frequencies at middle latitudes by MIPAS is robust.

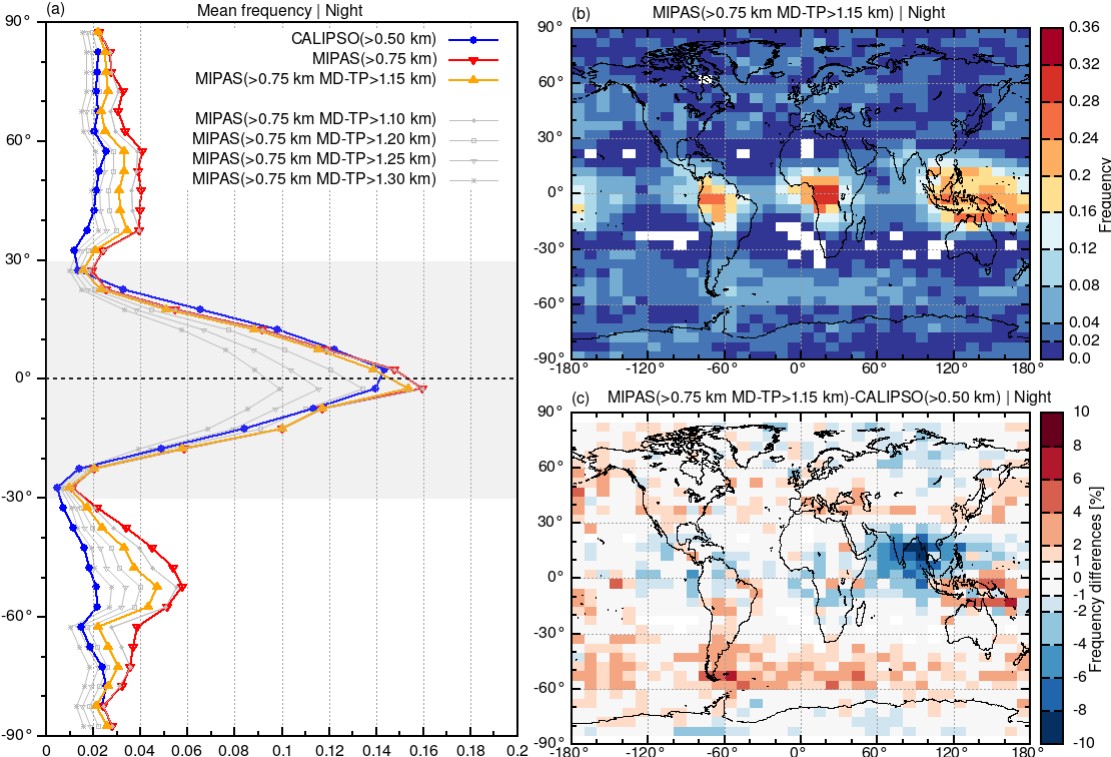

**Figure 10.** Sensitivity test on MIPAS stratospheric cirrus cloud detections applying an additional criterion regarding the mean distance (MD-TP) of the CTHs to the tropopause for each grid box (see text for details). Plots show a) zonal mean CTH occurrence frequencies, b) geospatial distribution of CTH occurrence frequencies, and c) difference between MIPAS and CALIPSO nighttime occurrence frequencies.

## 5 Comparison to previous stratospheric cirrus cloud statistics derived from CALIPSO

The CALIPSO level 2 V4.x data product used in this study was significantly improved with respect to the aerosol and cloud classification (Liu et al., 2019) and the cloud detection sensitivity by applying more accurate calibration algorithms, higher lidar ratios, and lower attenuated backscatter coefficients (Kar et al., 2018; Vaughan et al., 2019; Young et al., 2018) compared to the CALIPSO V3 data product that was used by Pan and Munchak (2011). To investigate the impact of these improvements, we analyzed the distribution of CTHs with respect to the tropopause for the same four years of CALIPSO measurements

from June 2006 to May 2010, the same stratospheric cirrus cloud definition (0.5 km above the local tropopause, and the same





latitude-longitude grid as in Pan and Munchak (2011). But different from Pan and Munchak (2011), we applied a PSC filter for polar winter conditions.

The geospatial distribution of stratospheric cirrus clouds, shown in Fig. 11 using the same $2° \times 3°$ latitude-longitude grid as Fig. 7 in Pan and Munchak (2011), exhibits similar patterns with the highest CTH frequencies of stratospheric cirrus clouds

in the tropics, but with larger absolute values in our study. At middle latitudes, more grid points with frequencies of 4-8 % are found over the northern Pacific Ocean, the northern Atlantic Ocean, northern Asian, the southern Atlantic and the southern Indian Ocean in our study (Fig. 11). At high latitudes (>60°) during polar winter both, our study and Pan and Munchak (2011), show enhanced CTH frequencies, but the occurrence frequencies by Pan and Munchak (2011) are significantly larger, reaching up to 24 % compared to up to ~8 % in our study. This difference we attribute to the PSC filtering that was applied in our study.

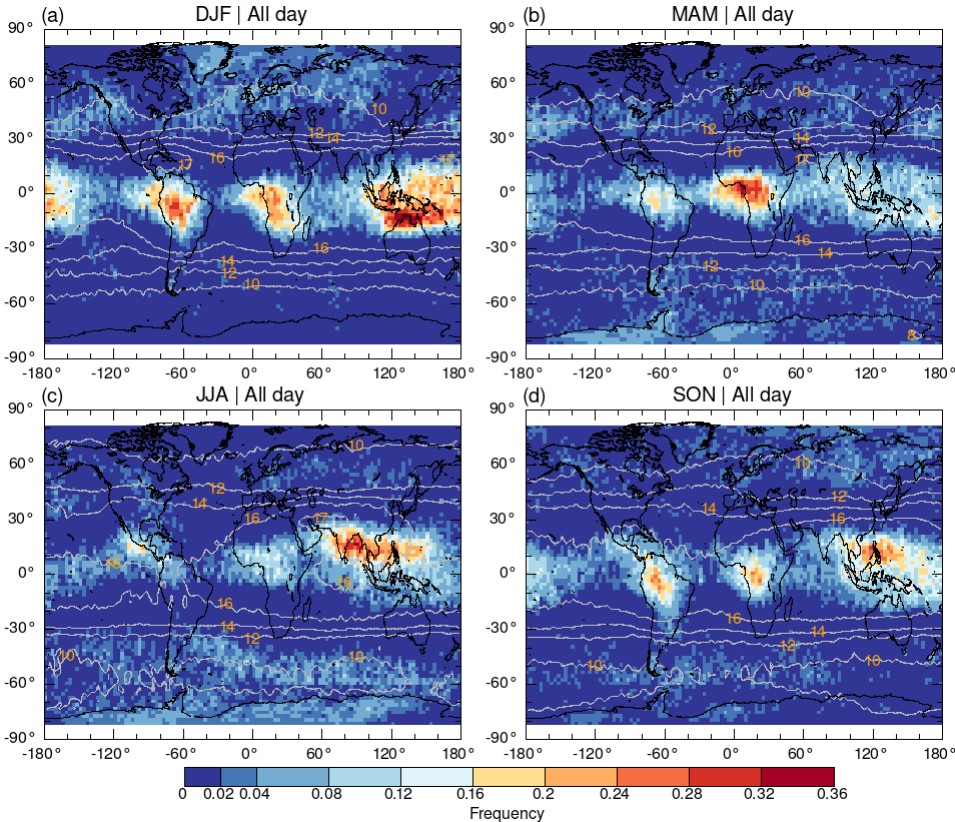

**Figure 11.** All day (average of day- and nighttime) seasonal CTH occurrence frequencies of stratospheric cirrus clouds derived from CALIPSO measurements between June 2006 and May 2010 for comparison with results of Pan and Munchak (2011). The maps are gridded on a 2°×3° latitude-longitude grid. The grey contour lines indicate mean tropopause heights.


The seasonally resolved vertical distribution of cirrus clouds around the tropopause, shown in Fig. 12, we compared with Fig. 10 in Pan and Munchak (2011). In both data sets, the maximum frequencies appear around the tropopause ($\pm 0.5$ km), the



highest CTH occurrence frequencies in the tropics are found in DJF, in NH middle latitudes also in DJF, and in SH middle
latitudes in JJA. However, in our study the occurrence frequencies are about 1 to 3 pp higher in the tropics and about 0.5 pp

higher in NH middle latitudes. Hence, using CALIPSO V4.x data and tropopauses derived from ERA-interim results in notably
larger CTH occurrence frequencies of stratospheric cirrus clouds than derived by Pan and Munchak (2011).

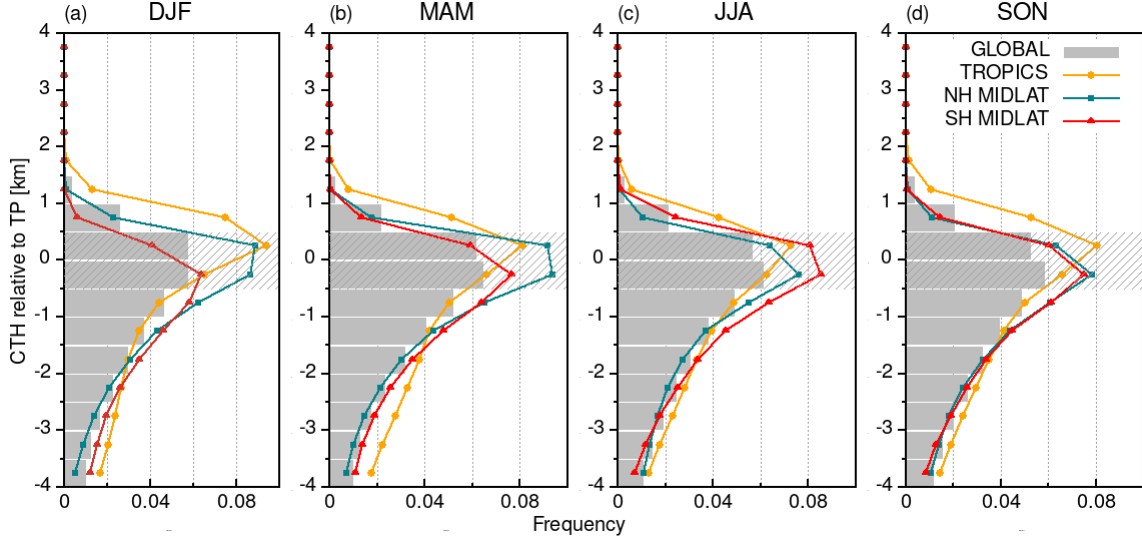

**Figure 12.** Vertical distributions of all day (average of day- and nighttime) CTHs relative to tropopause for four seasons a) DJF, b) MAM,
c) JJA, d) SON derived from CALIPSO observations between June 2006 and May 2010 for comparison with results of Pan and Munchak
(2011).

## 6 Conclusions

In this study, we derived global stratospheric cirrus clouds from the mid-infrared limb emission sounder MIPAS and the
CALIPSO lidar level 2 version 4.x data for the time period between June 2006 and April 2012 that is covered by both instru-

ments. The local tropopause heights for each satellite profile were derived from the ERA-Interim reanalysis using the WMO
criterion for the first thermal tropopause.

For CALIPSO, cirrus cloud top heights more than 0.5 km above the local tropopause were considered stratospheric. Due to
the better detection sensitivity of CALIPSO nighttime measurements, we only considered the nighttime measurements. The
highest CTH occurrence frequencies of stratospheric cirrus clouds were found in the tropics over the continents of Equatorial

Africa, South and Southeast Asia, and South America and the western Pacific warm pool. The hotspots follow the ITCZ and the
maximum occurrence frequencies reached more than 36 % in DJF. The zonal mean CTH occurrence frequency of stratospheric
cirrus in the tropics is about 7 %. A secondary, but much weaker, stratospheric cirrus cloud cluster is located in the middle
latitudes of both hemispheres with a zonal mean occurrence frequency of about 2 % and occurrence frequencies of up to 12 %.



Our findings qualitatively agree with the results by Pan and Munchak (2011), but are quantitatively higher. One reason for the

higher frequencies is that we looked at nighttime data only. In addition, the comparison of night and day averages for the same time period as investigated by Pan and Munchak (2011) showed that using the combination of CALIPSO V4.x data and ERA-interim causes higher occurrence frequencies e. g. reaching up to 36 % in several grid boxes in DJF compared to a maximum of 32 % in a single grid box in Pan and Munchak (2011), who used CALIPSO V3 data and GFS tropopauses.

The largest challenge for deriving stratospheric cirrus clouds from MIPAS data was its rather large field of view and the

vertical sampling of 1.5 km. Although MIPAS is known to overestimate cloud top heights of optically thin and thick clouds ($\tau$>0.03) by about 0.75 km in average (Sembhi et al., 2012; Griessbach et al., 2020), we did not find an obvious altitude offset when comparing MIPAS and CALIPSO cloud occurrence frequencies relative to the tropopause (Fig. 5). But, MIPAS systematically provided higher cloud occurrence frequencies than CALIPSO nighttime measurements. The overall higher detection frequencies we attributed to MIPAS larger sampling volume at the tangent point and the higher detection sensitivity

reaching down to $\tau$ of $10^{-5}$ compared to $10^{-3}$ for CALIPSO.

However, to make sure we did not overestimate cloud top heights, especially in the middle latitudes, we scaled the MIPAS stratospheric CTH occurrence frequencies in the tropics to CALIPSO. The minimum difference between MIPAS and CALIPSO we yield for MIPAS data more than 0.75 km above the tropopause. While the overall patterns and and the average occurrence frequency in the tropics agreed, we found about two to three times more stratospheric cirrus clouds (up to 6 %) in the middle and

high latitudes than for CALIPSO (up to 2.5 %). In a further sensitivity test to exclude sampling artefacts of MIPAS changing tangent heights with latitude, we investigated the mean distance of the stratospheric cirrus clouds to the tropopause. For a mean distance of 1.15 km, we found the best agreement with CALIPSO in the tropics. Since the mean distance to the tropopause is larger than for the 0.75 km-above-the-tropopause criterion the number of stratospheric cirrus clouds at middle and high latitudes became smaller (up to 4 %), but still by a factor of 2 larger than for CALIPSO. The CTH occurrence frequencies of stratospheric

cirrus clouds derived from MIPAS are closer to the occurrence frequencies of about 5 to 7 % found in previous studies at middle latitudes (Clodman, 1957; Goldfarb et al., 2001; Spang et al., 2015). Although we cannot definitely quantify the occurrence frequencies from MIPAS, we conclude that more stratospheric cirrus clouds are present and that they are optically thin, too thin to be detected by CALIPSO.

The comparison of MIPAS daytime and nighttime measurements showed slightly higher occurrence frequencies in the

tropics during nighttime of about 1 pp in zonal mean and slightly lower occurrence frequencies at middle latitudes of about 0.5 pp in zonal mean (Fig. 8). This result is in line with other observations of high altitude cirrus clouds that show little diurnal cycle and thin cirrus in particular showing no obvious diurnal pattern (Wylie et al., 1994). The comparison of CALIPSO daytime and nighttime stratospheric cirrus cloud occurrence frequencies shows significantly higher occurrence frequencies in the tropics of 10 % during nighttime compared to 4 % during daytime. At middle latitudes the occurrence frequencies also

differ by a factor of 2 with 2 % at nighttime and 1 % at nighttime (Fig. 2). This difference is due to the different detection sensitivities between CALIPSO daytime and nighttime measurements. From this we conclude that stratospheric cirrus clouds are optically thin and for this type of clouds CALIPSO operates at its detection limit.

We revisited the global stratospheric cirrus clouds with high vertical resolution and high detection sensitivity satellite observations in this work. More stratospheric cirrus clouds were detected in middle latitudes with higher detection sensitivity
measurements. Future work will have to assess the impact of these optically thin cirrus clouds on the radiative budget and climate. Furthermore, the individual characteristics of a single satellite sensor, i. e., its detection sensitivity and spatiotemporal coverage and resolution, may still pose limitations for the results. Future work using both, high resolution and high detection sensitivity measurements, or combining different measurement techniques will push forward a better understanding of the characteristics and distributions of stratospheric cirrus clouds on a global scale.

*Data availability.* Stratospheric cirrus cloud top heights from CALIPSO and MIPAS are available upon request from the contact author, Ling Zou (l.zou@fz-juelich.de; cheryl_zou@whu.edu.cn). MIPAS cloud data including aerosol and ice cloud flags are available at https: //datapub.fz-juelich.de/slcs/mipas/aerosol_clouds/index.html (last access: 25 March 2020). Tropopause data are available at https://www. re3data.org/repository/r3d100013201 (last access: 25 March 2020).

*Author contributions.* LZ, SG, and LH conceived the study design. LZ conducted the formal analysis and compiled the results. SG provided
the MIPAS data. LH provided the ERA-Interim tropopause data. BG and LCW supported the CALIPSO data processing. LZ wrote the manuscript with contributions from all co-authors.

*Competing interests.* The authors declare that they have no conflict of interest.

*Acknowledgements.* This work was supported by the National Natural Science Foundation of China under grant No. 41801021 and the International Postdoctoral Exchange Fellowship Program 2018 under grant No. 20181010. CALIPSO data were obtained from the NASA
Langley Research Center Atmospheric Science Data Center. The MIPAS data were provided by the European Space Agency. The ERA-Interim reanalysis data were obtained from the European Centre for Medium-Range Weather Forecasts. We gratefully acknowledge the computing time granted on the supercomputers JURECA and JUWELS at Forschungszentrum Jülich. We also would like to thank Dr. Reinhold Spang from the Forschungszentrum Jülich for useful discussions.





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
