# Peer review of "Revisiting global satellite observations of stratospheric cirrus clouds"

_Atmospheric Chemistry and Physics, 2020_

## Referee Comment (RC1) · Anonymous Referee #2 · 23 May 2020

General Comments:

This carefully written manuscript advances our global/seasonal estimates of the occurrence frequency of cirrus clouds that, for CALIPSO satellite measurements, reside 0.5 km or higher above the tropopause; that is, stratospheric cirrus clouds. Polar stratospheric clouds or PSCs, primarily occurring over Antarctica, have been filtered out of this data set, and are not considered. This study builds on other studies by using for the first time the Michelson Interferometer for Passive Atmospheric Sounding (MIPAS; onboard ESA's Envisat) limb measurements to detect stratospheric cirrus clouds 0.75 km above the tropopause (identified using ERA-interim global reanalysis). Due to the larger sample volume of the MIPAS atmospheric limb measurements, MIPAS can detect cirrus clouds down to optical depths of $2 \times 10^{-5}$ compared to a detection threshold

for CALIPSO of 1 x 10-3, resulting in 2 to 3 times more stratospheric cirrus cloud occurrence frequencies outside the tropics relative to CALIPSO.

I have only six fairly minor comments listed below. The paper is very well written and organized and makes a significant contribution to our knowledge of stratospheric cirrus clouds. The scientific methods and assumptions are valid and clearly outlined, and the results appear traceable.

Major and Minor Comments:

1. Line 23: Not all cirrus clouds are optically thin; suggest removing "optically thin".

2. Lines 31-37: Cirrus cloud occurrence frequencies vary considerably partly due to satellite instrument sensitivity (as mentioned here), but this may also depend on how cirrus clouds are defined (e.g. in terms of a temperature or altitude threshold, or in terms of pressure and optical depth as with ISCCP, etc.). Please indicate the cirrus definition when citing estimates of occurrence frequency for cirrus clouds.

3. Lines 123-133: How is ACI calculated? Is it simply the largest of the two values [i.e. ACI=max(CI, AI)]?

4. Lines 192-194: The JJA band of enhanced (4-8%) stratospheric cirrus downwind of the southern tip of South America is a "hotspot" for mountain-induced stratospheric gravity wave formation as described by Jiang et al. (2002, JGR) and Hoffmann et al. (2016, ACP), although the maximum in gravity waves in Jiang et al. is during SON (rather than JJA as shown in Hoffmann et al.). This suggests that the observed stratospheric cirrus enhancement in Fig. 3c resulted from mountain-induced gravity waves. The occurrence frequency for relatively thick ($0.3 < \tau < 3.0$) cirrus clouds downwind of the Southern Andes was found to be greatest during JJA in Mitchell et al. (2016, ACPD; see Mitchell et al. 2018, ACP, for additional details), corroborating the results of this study somewhat.

5. Lines 388-390: Please indicate that this sensitivity refers to cloud optical depth.

6. Line 410: Suspect typo: "1% at nighttime" => "1% during daytime"?

Please also note the supplement to this comment:
https://www.atmos-chem-phys-discuss.net/acp-2020-304/acp-2020-304-RC1-supplement.pdf
* * *

---

## Author Comment (AC1) · 8 Jun 2020

**Reply to review comments**

We express our gratitude for the time and effort dedicated to the reviewing of our manuscript. We considered all comments and provided our detailed point-by-point responses below.

**Anonymous Referee #2**

**General Comments**

This carefully written manuscript advances our global/seasonal estimates of the occurrence frequency of cirrus clouds that, for CALIPSO satellite measurements, reside 0.5 km or higher above the tropopause; that is, stratospheric cirrus clouds. Polar stratospheric clouds or PSCs, primarily occurring over Antarctica, have been filtered out of this data set, and are not considered. This study builds on other studies by using for the first time the Michelson Interferometer for Passive Atmospheric Sounding (MIPAS; onboard ESA's Envisat) limb measurements to detect stratospheric cirrus clouds 0.75 km above the tropopause (identified using ERA-interim global reanalysis). Due to the larger sample volume of the MIPAS atmospheric limb measurements, MIPAS can detect cirrus clouds down to optical depths of 2 x 10-5 compared to a detection threshold for CALIPSO of 1 x 10-3, resulting in 2 to 3 times more stratospheric cirrus cloud occurrence frequencies outside the tropics relative to CALIPSO.

I have only six fairly minor comments listed below. The paper is very well written and organized and makes a significant contribution to our knowledge of stratospheric cirrus clouds. The scientific methods and assumptions are valid and clearly outlined, and the results appear traceable.

Answer: Thank you very much for your encouraging comments!

**Comment \* 1**

Line 23: Not all cirrus clouds are optically thin; suggest removing "optically thin".

**Answer: Thanks, done.**

"Cirrus clouds are ice clouds that form at cold temperatures in the middle and upper troposphere."

**Comment \* 2**

Lines 31-37: Cirrus cloud occurrence frequencies vary considerably partly due to satellite instrument sensitivity (as mentioned here), but this may also depend on how cirrus clouds are defined (e.g. in terms of a temperature or altitude threshold, or in terms of pressure and optical depth as with ISCCP, etc.). Please indicate the cirrus definition when citing estimates of occurrence frequency for cirrus clouds.

Answer: Exactly, we have added cirrus clouds definitions according to different references.

The sentence was revised to:

"Depending on the satellite instruments sensitivities and cirrus cloud definition, the derived occurrence frequencies significantly differ, e. g. in global average 34.9% cirrus clouds above 500 hPa were observed by the High Resolution Infrared Radiometer Sounder (HIRS) between June 1989 to May 1993 (Wylie et al., 1994), 16.7% cirrus clouds, with cloud top temperature below -40° C and a visible optical depth below  $\tau \approx 3.0$ , were derived from a joint analysis of the space-borne cloud radar (CloudSat) and the Cloud-Aerosol Lidar and Infrared Pathfinder Satellite Observations (CALIPSO) for the period from June 2006 to June 2007 (Sassen et al., 2008), and 13.5% cirrus clouds with cloud top pressure below 440 mb and an optical thickness below 3.6 were reported in the International Satellite Cloud Climatology Project (ISCCP) D2 data, that was acquired between 1984 and 2004 by nadir viewing satellite instruments (Eleftheratos et al., 2007)."

**Comment \* 3**

Lines 123-133: How is ACI calculated? Is it simply the largest of the two values [i.e. ACI=max(CI, AI)]?

**Answer:** Yes. The ACI is the maximum value of the cloud index (CI) and the aerosol index (AI). For clarity we added the formula ACI=max(CI, AI) to the manuscript.

**Comment \* 4**

Lines 192-194: The JJA band of enhanced (4-8%) stratospheric cirrus downwind of the southern tip of South America is a "hotspot" for mountain-induced stratospheric gravity wave formation as described by Jiang et al. (2002, JGR) and Hoffmann et al. (2016, ACP), although the maximum in gravity waves in Jiang et al. is during SON (rather than JJA as shown in Hoffmann et al.). This suggests that the observed stratospheric cirrus enhancement in Fig. 3c resulted from mountain-induced gravity waves. The occurrence frequency for relatively thick ( $0.3 < \tau < 3.0$ ) cirrus clouds downwind of the Southern Andes was found to be greatest during JJA in Mitchell et al. (2016, ACPD; see Mitchell et al. 2018, ACP, for additional details), corroborating the results of this study somewhat.

**Answer:** Thanks for those detailed information. We have added some discussion about the effects of deep convection and gravity waves on occurrences of stratospheric cirrus clouds at middle and high latitudes.

"Although the occurrence frequencies at middle latitudes are lower compared to the tropics, we see higher occurrence frequencies during the winter months. The stratospheric cirrus clouds at middle and high latitudes are located at and downwind of gravity wave hotspots (Hoffmann et al., 2013). In DJF, stratospheric cirrus clouds over North America, the northern hemisphere Atlantic, and Eurasia are correlated with orographically and convectively induced gravity wave hotspots, whereas the stratospheric cirrus clouds over the Northern Pacific are solely correlated with deep convection (Hoffmann et al., 2013). In JJA, stratospheric cirrus clouds occur in the oceanic downwind region of the southern tip of South America, which is a strong hotspot of orographic gravity waves (Jiang et al., 2002; Hoffmann et al., 2013). "

**Comment \* 5**

Lines 388-390: Please indicate that this sensitivity refers to cloud optical depth.

**Answer: Done.**

"The overall higher detection frequencies we attributed to MIPAS larger sampling volume at the tangent point and the higher detection sensitivity reaching down to cloud optical depths  $\tau$  of  $10^{-5}$  compared to  $10^{-3}$  for CALIPSO."

**Comment \* 6**

Line 410: Suspect typo: "1% at nighttime" =>"1% during daytime"?

**Answer: Thanks, done.**

"At middle latitudes the occurrence frequencies also differ by a factor of 2 with 2% at nighttime and 1% at daytime."

---

## Referee Comment (RC2) · Vincent Noel (Referee) · 24 Jun 2020

In this paper, the authors update what is known about the observed spatial, vertical, and seasonal distribution of stratospheric cirrus. They do so using refined retrievals from spaceborne lidar (CALIPSO) and spaceborne limb interferometer (MIPAS), and ERA5 reanalyses. They compare their results with previous climatologies from the literature.

I can only wish my first submissions were as clear in their purpose and as well-written as this article. The structure is focused and to the point. The methodology is sound and careful. The figures are all clear and informative, they convey their message well. The results are convincing and bring an updated climatology of stratospheric cirrus.

[Figure]

The appropriate literature (that I know of) is referenced and compared against. This is a very good paper that should be published. I have a few minor comments below.

The only negative I can see with the paper is that it puts the CALIPSO results first, so at first glance it "merely" looks like it is confirming already known facts (ie Pan and Munchak's climatologies), with updated reanalyses and retrievals. I find the CALIPSO results are the less interesting – unless I'm mistaken Section 5 shows very little improvement in stratospheric cirrus detection or improved understanding compared to P&M. The MIPAS results appear to me much more interesting: showing that the cover of stratospheric cirrus is double the one previously thought is an important result. Unless I'm mistaken, it is the first time that MIPAS stratospheric cirrus are presented. I feel your work would be better served if the paper led with the fully-new MIPAS results and put them in perspective against the already-known CALIPSO detections.

**Minor comments**

1. L. 28: "the characteristics and distribution of cirrus clouds are among the most sensitive parameters to climate variability" – would you have a reference for that? 2. I had noticed the enhanced frequency of stratospheric cirrus in the southern ocean in JJA (fig. 3c), but I see that you've already covered that with the first reviewer.

3. L. 269-273: If I understand correctly, MIPAS underestimates the amount of stratospheric clouds above the Bay of Bengal compared to CALIPSO (blue spot in Fig. 6c), while it finds more of them everywhere else. You relate this underestimate with the Asian monsoon, but could you propose an explanation why the Asian monsoon would lead to less stratospheric cirrus seen by MIPAS above the Bay of Bengal? This location is suspiciously close to the Asian Tropopause Aerosol Layer, which is also related to the Asian monsoon and also maximum in JJA (cf. Vernier et al. 2011 and Thomason and Vernier, 2013). Could the presence of aerosols in this region near the Tropopause influence cirrus retrievals of CALIPSO or MIPAS?

4. L. 326-330: You mention that in Dauhut et al. (2020) CATS measurements were

derived from 5-km along-track averages in both daytime and nighttime conditions, in contrast with Noel et al. (2018) in which daytime measurements were averaged horizontally over 60 km to align with the nighttime sensitivity. In truth, both papers used the same detection algorithm, which apparently is not described well: for both daytime and nighttime data, layers are first detected at 60-km horizontal averaging, then at 5-km horizontal averaging. If a layer is detected at both resolutions, the heights for 5-km averaging are used, otherwise 60-km. This implies that, in general, daytime detections occur at 60-km horizontal averaging and nighttime detections at 5-km averaging. In the cloud product, all detections are reported on a 5-km horizontal grid. For your purposes, I guess the discrepancy between Dauhut et al. and your results can still be attributed to changes in CATS detection sensitivity with incoming solar pollution, which are not perfectly understood yet.

References

Vernier, J.-P., L. W. Thomason, and J. Kar (2011), CALIPSO detection of an Asian tropopause aerosol layer, Geophys. Res. Lett., 38, L07804, doi:10.1029/2010GL046614.

Thomason, L. W., and J.-P. Vernier (2013), Improved SAGE II cloud/aerosol categorization and observations of the Asian tropopause aerosol layer: 1989–2005, Atmos. Chem. Phys., 13, 4605–4616, doi:10.5194/acp-13-4605-2013.

---

## Author Comment (AC2) · 10 Jul 2020

**Reply to review comments**

We express our gratitude for the time and effort dedicated to the reviewing of our manuscript. We considered all comments and provided our detailed point-by-point responses below.

**Reviewer #1**

**General Comments**

In this paper, the authors update what is known about the observed spatial, vertical, and seasonal distribution of stratospheric cirrus. They do so using refined retrievals from spaceborne lidar (CALIPSO) and spaceborne limb interferometer (MIPAS), and ERA5 reanalyses. They compare their results with previous climatologies from the literature.

I can only wish my first submissions were as clear in their purpose and as well-written as this article. The structure is focused and to the point. The methodology is sound and careful. The figures are all clear and informative, they convey their message well. The results are convincing and bring an updated climatology of stratospheric cirrus.

The appropriate literature (that I know of) is referenced and compared against. This is a very good paper that should be published. I have a few minor comments below.

The only negative I can see with the paper is that it puts the CALIPSO results first, so at first glance it "merely" looks like it is confirming already known facts (ie Pan and Munchak's climatologies), with updated reanalyses and retrievals. I find the CALIPSO results are the less interesting – unless I'm mistaken Section 5 shows very little improvement in stratospheric cirrus detection or improved understanding compared to P&M. The MIPAS results appear to me much more interesting: showing that the cover of stratospheric cirrus is double the one previously thought is an important result. Unless I'm mistaken, it is the first time that MIPAS stratospheric cirrus are presented. I feel your work would be better served if the paper led with the fully-new MIPAS results and put them in perspective against the already-known CALIPSO detections.

**Answer:** Thank your very much for appreciating our work.

Yes, this is the first time that MIPAS measurements are used to derive stratospheric cirrus clouds. Thanks for your suggestion on changing the order of presenting the results for the two satellite instruments. We actually have discussed a lot about it. Finally, we structured the manuscript as it is now, because the numbers derived from MIPAS rely on CALIPSO data that was used to scale the MIPAS results in tropics. We think presenting CALIPSO results first makes our approach of deriving stratospheric cirrus cloud occurrence frequencies from MIPAS data better comprehensible. But surely, the results from MIPAS would be more interesting to readers. Hence, we made some adjustments to abstract to emphasize the MIPAS results. The revised abstract reads now:

"As knowledge about the cirrus clouds in the lower stratosphere is limited, reliable long-term measurements are needed to assess their characteristics, radiative impact and important role in upper troposphere and lower stratosphere (UTLS) chemistry. We used six years (2006 – 2012) of Michelson Interferometer for Passive Atmospheric Sounding (MIPAS) measurements to investigate the global and seasonal distribution of stratospheric cirrus clouds and compared the MIPAS results with results derived from the latest version (V4.x) of the Cloud-Aerosol Lidar and Infrared Pathfinder Satellite Observations (CALIPSO) data. For the identification of stratospheric cirrus clouds, precise information on both, the cloud top height (CTH) and the tropopause height is crucial. Here, we used lapse rate tropopause heights estimated from the ERA-Interim global reanalysis.

Considering the uncertainties of the tropopause heights and the vertical sampling grid, we defined CTHs more than 0.5 km above the tropopause as stratospheric for CALIPSO data. For MIPAS data, we took into account the coarser vertical sampling grid and the broad field of view, so that we considered cirrus CTHs detected more than 0.75 km above the tropopause as stratospheric. Further sensitivity tests were conducted to rule out sampling artefacts in MIPAS stratospheric.

The global distribution of stratospheric cirrus clouds was derived from nighttime measurements because of the higher detection sensitivity of CALIPSO. In both data sets, MIPAS and CALIPSO, the stratospheric cirrus cloud occurrence frequencies are significantly higher in the tropics than in the extra-tropics. The tropical hotspots of stratospheric cirrus clouds are associated with deep convection and are located over Equatorial Africa, South and Southeast Asia, the western Pacific and South America. Stratospheric cirrus clouds were more often detected in December-February (15 %) than June-August (8 %) in the tropics ($\pm 20°$). At northern and southern middle latitudes (40-60°), MIPAS observed about twice as much stratospheric cirrus clouds as CALIPSO (occurrence frequencies of $4 - 5$ % for MIPAS compared to about 2 % for CALIPSO). We attribute the more frequent observations of stratospheric cirrus clouds with MIPAS to its higher detection sensitivity to optically thin clouds.

In contrast to the difference between daytime and nighttime occurrence frequencies of stratospheric cirrus clouds by a factor of about 2 in zonal mean in the tropics (4 % and 10 %, respectively) and middle latitudes for CALIPSO data, there is little diurnal cycle in MIPAS data, in which the difference of occurrence frequencies in the tropics is about 1 percentage point in zonal mean and about 0.5 percentage point at middle latitudes. The

difference between CALIPSO day and night measurements can also be attributed to their differences in detection sensitivity.

Future work should focus on better understanding the origin of the stratospheric cirrus clouds and their impact on radiative forcing and climate."

**Comment * 1**

L. 28: "the characteristics and distribution of cirrus clouds are among the most sensitive parameters to climate variability" – would you have a reference for that?

**Answer:** Done, related references have been added.

"The characteristics and distribution of cirrus clouds are among the most sensitive parameters to climate variability (Muri et al.,2014; Kärcher,2018). "

Muri, H., Kristjánsson, J. E., Storelvmo, T., and Pfeffer, M. A.: The climatic effects of modifying cirrus clouds in a climate engineering framework, Journal of Geophysical Research: Atmospheres, 119, 4174–4191, https://doi.org/10.1002/2013JD021063, 2014.

Kärcher, B.: Formation and radiative forcing of contrail cirrus, Nature Communications, 9, https://doi.org/10.1038/s41467-018-04068-0, 2018

**Comment * 2**

I had noticed the enhanced frequency of stratospheric cirrus in the southern ocean in JJA (fig. 3c), but I see that you've already covered that with the first reviewer.

**Answer:** Thanks. Our response is also presented below.

"Although the occurrence frequencies at middle latitudes are lower compared to the tropics, we see higher occurrence frequencies during the winter months. The stratospheric cirrus clouds at middle and high latitudes are located at and downwind of gravity wave hotspots (Hoffmann et al., 2013). In DJF, stratospheric cirrus clouds over North America, the northern hemisphere Atlantic, and Eurasia are correlated with orographically and convectively induced gravity wave hotspots, whereas the stratospheric cirrus clouds over the Northern Pacific are solely correlated with deep convection (Hoffmann et al., 2013). In JJA, stratospheric cirrus clouds occur in the oceanic downwind region of the southern tip of South America, which is a strong hotspot of orographic gravity waves (Jiang et al., 2002; Hoffmann et al., 2013). "

**Comment * 3**

L. 269-273: If I understand correctly, MIPAS underestimates the amount of stratospheric clouds above the Bay of Bengal compared to CALIPSO (blue spot in Fig. 6c), while it finds more of them everywhere else. You relate this underestimate with the Asian monsoon, but could you propose an explanation why the Asian monsoon would lead to less stratospheric cirrus seen by MIPAS above the Bay of Bengal? This location is suspiciously close to the Asian Tropopause Aerosol Layer, which is also related to the Asian monsoon and also maximum in JJA (cf. Vernier et al. 2011 and Thomason and Vernier, 2013). Could the presence of aerosols in this region near the Tropopause influence cirrus retrievals of CALIPSO or MIPAS?

**Answer:** Yes, CALIPSO detected more stratospheric cirrus clouds over the Bay of Bengal in boreal summer (JJA). We considered it would be related to deep convection during the Asian monsoon season. But we could not identify the exact reason for now. We ruled out an effect of the Asian Tropopause Aerosol Layer (ATAL): 1) The ATAL is usually detected between 15 - 17 km, but the mean tropopause over this region in JJA is higher than 17 km. 2) For MIPAS we filtered out aerosol, but in the MIPAS aerosol data we did not find an enhanced aerosol layer over the Asian monsoon region. 3) To make the actually very thin ATAL visible in CALIPSO data, multiple profiles have to be averaged (over several weeks and a larger region) to obtain a sufficient signal-to-noise ratio. In this study we used the CALIPSO vertical feature mask data that is not sensitive to the ATAL.

Further we thought that in regions with deep convection, where we expect horizontal cloud inhomogeneities, the different sampling volumes might play a role. We added the following description of a further sensitivity test

that we performed:

"As different sampling volumes in MIPAS and CALIPSO may produce uncertainties, we calculated the fraction of stratospheric cirrus clouds in UTLS clouds (tropopause $\pm 4\,\mathrm{km}$) instead of in all profiles. This way a potential uncertainty due to the sampling volume is present in the nominator and denominator and hence should cancel out. While the absolute number of occurrence frequencies of stratospheric cirrus clouds in UTLS clouds increases compared to the occurrence frequencies of stratospheric cirrus clouds in all profiles, the factor between MIPAS and CALIPSO stratospheric cloud occurrence frequencies at middle latitudes remains the same, indicating that our result is robust and the different sampling volumes do not impair our results. Moreover, tropical cirrus layers near the tropopause extend horizontally over hundreds to thousand kilometers (Winker et al., 1998) and over half the horizontal scales of cirrus clouds at $16$‑$17\,\mathrm{km}$ altitudes are larger than $100\,\mathrm{km}$ (Massie et al., 2010). Due to the large horizontal scale of tropopause layer cirrus clouds, the effect of the sampling volume on the detection of CTH occurrence frequencies with MIPAS and CALIPSO would be negligible."

Finally, it may be related to different measurement times between the two satellites. This is a quite interesting point for future work.

**Comment ∗ 4**

L. 326-330: You mention that in Dauhut et al. (2020) CATS measurements were derived from 5-km along-track averages in both daytime and nighttime conditions, in contrast with Noel et al. (2018) in which daytime measurements were averaged horizontally over 60 km to align with the nighttime sensitivity. In truth, both papers used the same detection algorithm, which apparently is not described well: for both daytime and nighttime data, layers are first detected at 60-km horizontal averaging, then at 5-km horizontal averaging. If a layer is detected at both resolutions, the heights for 5-km averaging are used, otherwise 60-km. This implies that, in general, daytime detections occur at 60-km horizontal averaging and nighttime detections at 5-km averaging. In the cloud product, all detections are reported on a 5-km horizontal grid. For your purposes, I guess the discrepancy between Dauhut et al. and your results can still be attributed to changes in CATS detection sensitivity with incoming solar pollution, which are not perfectly understood yet.

**Answer:** Thank you for detailed explanation of CATS cloud detection algorithm. We have deleted those related inappropriate descriptions in our manuscript.

"A recent study on stratospheric cirrus cloud occurrences in the tropics derived from CATS measurements reports differences of about 3 to $10\,\mathrm{pp}$ in DJF and 5 to $7\,\mathrm{pp}$ in JJA between $10\,\mathrm{am}$ and $10\,\mathrm{pm}$ (Dauhut et al.,2020). This differs from our results that show only $1\,\mathrm{pp}$ difference between $10\,\mathrm{am}$ and $10\,\mathrm{pm}$ measurements. As the detection sensitivity of CATS measurements averaged over $5\,\mathrm{km}$ during daytime is about 1.5 orders of magnitude lower than during nighttime (Yorks et al.,2016), we consider the different detection sensitivities of CATS daytime and nighttime measurements as the main cause for the differences."